# Electrophysiological properties and heart rate variability of patients with thalassemia major in Jakarta, Indonesia

**Rubiana Sukardi** [1]* , **Pustika Amalia Wahidiyat**[2] , **Phebe Anggita Gultom**[3] ,
**Mokhammad Ikhsan**[4] , **Muhammad Yamin**[5] , **Simon Salim**[5] , **Mulyadi M. Djer**[4]

**1** Integrated Cardiovascular Service, Univeristy of Indonesia, Cipto Mangunkusumo General Hospital, Central Jakarta, Indonesia, **2** Department of Child Health, Division of Hematology-Oncology, University of Indonesia, Cipto Mangunkusumo General Hospital, Central Jakarta, Indonesia, **3** Faculty of Medicine University of Indonesia, Central Jakarta, Indonesia, **4** Department of Child Health, Division of Cardiology, University of Indonesia, Cipto Mangunkusumo General Hospital, Central Jakarta, Indonesia, **5** Department of Internal Medicine, Division of Cardiology, University of Indonesia, Cipto Mangunkusumo General Hospital, Central Jakarta, Indonesia

☯ These authors contributed equally to this work.
* rubiana_sukardi@yahoo.com

**Data Availability Statement:** All relevant data are contained within the manuscript and its Supporting Information files.

## Abstract

Beta thalassemia major (TM) is a common hereditary disease in Indonesia. Iron overload due to regular transfusion may induce myocardial iron deposition leading to electrophysiological dysfunction and functional disorders of the heart. Ventricular arrhythmia is one of the most common causes of sudden cardiac death in thalassemia patients. This cross-sectional study of 62 TM patients aged 10–32 years in Cipto Mangunkusumo General Hospital was done to assess their electrophysiological properties and heart rate variability, including 24-hour Holter monitoring, signal averaged electrocardiogram (SAECG) for detection of ventricular late potential (VLP), and determination of heart rate variability (HRV). We also assessed their 12-lead ECG parameters, such as P wave, QRS complex, QT/ QTc interval, QRS dispersion, and QT/ QTc dispersion. Iron overload was defined by T2-star magnetic resonance (MR-T2*) values of less than 20 ms or ferritin level greater than 2500 ng/mL. Subjects were grouped accordingly. There were significant differences of QTc dispersion (p = 0.026) and deceleration capacity (p = 0.007) between MR-T2* groups. Multivariate analysis showed an inverse correlation between QTc dispersion and MR-T2* values. There was a proportional correlation between heart rate deceleration capacity in the low MR-T2* group (p = 0.058) and the high ferritin group (p = 0.007). No VLPs were detectable in any patients. In conclusion, prolonged QTc dispersion and decreased heart rate deceleration capacity were significantly correlated with greater odds of iron overload among patients with Thalassemia major.

## Introduction

Beta thalassemia major (TM) is one of the most common occurrences in hereditary hemoglobinopathy, due to deficiency or absence of β-globin subunits, which leads to ineffective

**Funding:** This research was supported by a Universitas Indonesia Research Grant, with internationally indexed [grant number NKB-1574/UN2.RST/HKP.05.00/2020], received by PAW. The funders had no role in study design, data collection and analysis, decision to publish, or preparation of the manuscript.

**Competing interests:** The authors have declared that no competing interests exist.

erythropoiesis. Diagnosis of TM is usually determined at the age of 1–2 years [1, 2]. Lifelong blood transfusions, along with extravascular hemolysis and increased intestinal absorption of iron contribute to iron overload in such patients. Consequently, iron chelation therapy is of utmost importance. However, chelation may not be sufficient to prevent overload and iron-related toxicity. The heart is one of the main organs affected by iron toxicity [3, 4]. The proposed mechanisms of TM-related cardiomyopathy include membrane damage by lipid peroxidase, mitochondrial respiratory chain enzyme damage, electrical activity disruptions, cardiac fibrosis, and gene expression alterations. These mechanisms can change both cardiac rhythm and structures [5, 6].

The two most common electrophysiological abnormalities in TM patients are disturbances in cardiac autonomic function and paroxysmal supraventricular tachyarrhythmias such as atrial fibrillation, atrial flutter and intra atrial reentrant tachycardia, while ventricular arrhythmias are manifestation that are more specific due to iron cardiotoxicity [7]. Conventional electrocardiography (ECG) has limited role to assess cardiac autonomic function and ventricular arrhythmia since the majority of patients with asymptomatic cardiomyopathy have normal ECG. Although T2-star magnetic resonance (MR-T2$^*$) has been demonstrated as promising for early diagnosis of iron overload cardiomyopathy, it cannot be used to evaluate cardiac autonomic disturbances or ventricular arrhythmias. Furthermore, its use is also limited by lack of availability and high cost, particularly in low- and middle-income countries.

Recently, Holter monitoring and signal averaged electrocardiogram (SAECG) have been used to determine cardiac autonomic function and ventricular late potential as predictors of ventricular arrhythmias in patients with post-myocardial infarction and heart failure [8, 9]. Despite their clinical significance, there have been a limited number of cross-sectional studies investigating Holter monitoring and SAECG in TM patients [4, 10]. As such, we aimed to compare electrophysiological properties of TM patients with and without iron overload at Cipto Mangunkusumo General Hospital, Jakarta using 24-hour Holter monitoring and SAECG measurements.

## Materials and methods

### Study protocol

We conducted a cross-sectional study involving 62 TM patients aged 10–32 years at the Pediatric Hematology outpatient clinic in Cipto Mangunkusumo General Hospital, Jakarta. Subjects received regular blood transfusions and various types of iron chelation. Patients with heart failure, acute or chronic inflammatory diseases, metabolic disorders, and those taking anti-arrhythmic and/or cardiovascular medication affecting sympatho-vagal balance in the past 3 months were excluded. The study protocol was reviewed and approved by the institutional ethics committee of the Faculty of Medicine, Universitas Indonesia (No. 19-11-1274). Written informed consent was obtained from all participants or their guardians in the study.

Subjects underwent history-taking, physical examinations, and blood tests. We collected demographic data as well as information on age at diagnosis, drug(s) usage, and blood transfusion frequency. We routinely measured blood pressure and heart rate measurements. The most recent ferritin level and MR-T2$^*$ data were evaluated within three months prior to data collection from medical records. Subsequently, patients were grouped according to their ferritin levels as well as their MR-T2$^*$ values. Cardiac T2$^*$ values of $\geq$ 20 ms were considered normal. At the time of entry into the study, a 24-hour Holter ECG recording for HRV analysis and SAECG for ventricular late potential were collected.

### A 24-hour Holter monitoring assessment

**Conduction pattern assessment.** We used the Vasomedical-BIOX[TM] 1306 ultra-compact 12-channel ECG (25 mm/second, amplitude of 10 mm/mV) with silver chloride electrodes. ECG waves were measured using EP Calipers application for Mac. Recordings were done at rest in a quiet room. Parameters used in this study were P wave duration, QRS duration, PR interval, QRS dispersion, QT interval, QTc interval, QT dispersion, and QTc dispersion. P wave interval was measured from the beginning to the end of the P wave. PR interval was measured from the beginning of the P wave to the beginning of Q wave. QRS duration was measured from the beginning of the Q wave to the end of the S wave. QT interval was measured from the beginning of the QRS complex until the end of the T wave, and the end of the T wave was determined by the tangent method. QT interval was corrected for heart rate using Bazett formula ($QTc = QT/\sqrt{R\text{-}R \text{ interval}}$) [11]. QRS, QT, and QTc dispersion were calculated by subtracting maximum durations of QRS, QT, and QTc in each lead with the minimum durations of QRS, QT, and QTc interval. All the parameters were measured from an average of three consecutive beats, mostly in lead II and V1. QRS, QT, and QTc dispersion were calculated manually. Heart rate deceleration capacity was defined as the expression of extrinsic neural control of the heart to decelerate the rate and measured quantitively using a 24-hour ambulatory electrocardiogram [12, 13].

**Heart rate variability.** Heart rate variability was measured using Vasomedical-BIOX[TM] 1306 ultra compact 12-channel ECG 24- hour Holter Recorder. We assessed the following parameters: standard deviation of all normal sinus RR intervals for 24-hour (SDNN), standard deviation of the averaged normal sinus RR intervals in all 5-minute segments (SDANN), root mean square of each successful RR interval difference (rMSSD), the percentage of successive normal sinus RR intervals >50 ms (pNN50), and the percentage of pNN50 [14]. Patients were asked to maintain their daily routine activities during the recording.

**Presence of ectopic beat.** The presence of ectopic beat, such as premature atrial contraction and ventricular premature beat, was evaluated from a 24-hour Holter recording.

### SAECG

We performed Vasomedical BIOX[TM] 1303 ultra compact 3-channel ECG Holter Recorder for SAECG and silver chloride electrode. Recordings were accepted for analysis if two of the following criteria were met: number of beats averaged >250, and mean noise levels were <1.0 μV for 25 Hz filters or <0.7 μV for 40 Hz filters. SAECG parameters were considered abnormal if total QRS duration was >114 ms, duration of low amplitude signals lower than 40 μV (LAS40) were >38 ms, and root mean square voltage of signals in the last 40 ms of the high frequency QRS intervals (RMS40) were <20 μV [15]. An SAECG study was considered positive for VLPs if two of these three parameters were abnormal, according to previous recommendations [16]. During SAECG assessment, the patient was lying down for fifteen minutes in a quiet room and all metal accessories had been removed.

### Statistical analysis

We performed descriptive and analytical statistic using SPSS Statistics 23.0. Continuous data are presented as mean and standard deviation (SD), or median and interquartile range (IQR) values as appropriate. We used independent t-test for normal data distribution and Mann-Whitney test for heterogenous data distribution. A p-value less than 0.05 was statistically significant. Multivariate analysis was performed for parameters with p-values less than 0.2 using linier regression. All parameters were compared between ferritin status groups (< 2500 ng/mL and ≥2500 ng/mL) as well as between MR-T2* status groups (<20 ms and ≥20 ms).

## Results

### Demographic characteristics

Of 62 subjects, most were in the 16-20-year age group and male (S1 Table). Fifty-seven (91.9%) subjects had high levels of ferritin ($\geq$2500 ng/mL), with 19 (30.6%) of the subjects had MR-T2* values less than 20 ms. Most subjects received deferiprone (61.3%) three times a day for chelation Mean age at first transfusion was 10 months, and mean duration of transfusion was 16.8 years.

### Conduction pattern assessment

Forty-nine patients underwent 12-lead ECG (3 refused, 10 were out of reach). QRS duration was wider in the low ferritin group compared to the high ferritin group(85.5 ms vs. 73.4 ms, respectively, p = 0.045). QTc dispersion was also significantly higher in the low MR-T2* group than in the high MR-T2* group, (72 ms vs 53.2 ms, respectively, p = 0.026) (S2 Table) Although both results were statistically significant, there were no differences of cardiac manifestation found between two groups.

### Heart rate variability and presence of ectopic beat

On HRV analysis, only heart rate deceleration capacity was lower in the low MR-T2* group than in the high group, (4.1 vs 5.2, respectively, p = 0.007) although there were no important clinical manifestationdifference between two groups (S3 Table).

### SAECG

No difference in SAECG parameters were observed between ferritin level groups or MR-T2* groups (S4 Table).

### Multivariate analysis

Both QTc dispersion and heart rate deceleration capacity were associated with MR-T2* status (S5 Table). Patients with lower QTc dispersion had a greater likelihood of high MR-T2* level, as did those with higher heart rate deceleration capacity.

## Discussion

This study explored the relationship of electrophysiological properties and heart rate variability with iron overload in 62 patients with thalassemia major in Indonesia. The results indicate that prolonged QTc dispersion and decreased heart rate deceleration capacity were significantly correlated with greater odds of iron overload among TM patients.

Serum ferritin level is commonly used to monitor iron concentration in the body, but MR-T2* is still considered to be the gold standard to quantify iron concentration in myocardial tissue [17]. Lower MR-T2* value is a result of higher iron concentration in myocardial tissue. MR-T2* values < 20 ms indicate cardiac siderosis, while MR-T2* values < 10 ms indicate a higher risk of symptomatic heart failure and mortality [18]. Most of our subjects had high serum ferritin level. This finding might have been related to poor compliance to taking chelating drugs, subjective feeling towards the drugs, and lack of drug availability [19]. However, more than 60% of subjects had normal MR-T2* values, possibly because the majority of our patients received deferiprone. Based on previous studies, deferiprone is known to have specific action on cardiac siderosis and is superior to subcutaneous deferoxamine in preventing morbidity and mortality due to cardiac siderosis [20–23].

A 12-lead ECG has been used as one of early diagnostic tools for analyzing electrical conduction disturbances [20, 24]. QTc interval is reasonably specific to diagnose iron overload cardiomyopathy. Previous studies revealed that increasing iron stores in men correlated with prolonged QTc interval and may induce cardiac arrhythmias, including torsades de pointes, ventricular fibrillation, and sudden cardiac death [25, 26]. We found no significant differences in QT and QTc intervals between ferritin groups nor between MR-T2* groups. Heart rate- corrected QT intervals were considered prolonged if >450 ms for males and >460 ms for females [27]. Using these definitions, 12.2% of our patients had prolonged QTc. QTc prolongation was found to correlate with iron status (ferritin) and cardiac iron (MR-T2*) [25], but we found no such correlations between QTc and iron status in our study. This observation might have been because subjects' ferritin or iron status was obtained within three months prior to the study, therefore, the iron status obtained may not have reflected the current or recent condition of the patients.

QT dispersion (QTd) and QTc dispersion are also sensitive for predicting repolarization abnormality. QTc dispersion is one of the surrogate markers for ventricular repolarization homogeneity [28]. Previous studies showed that the prevalence of increased QTc dispersion > 65 ms increased the risk of ventricular arrhythmias [29, 30]. In our study, this condition was found in 31% of patients with high ferritin (vs 50% in the low ferritin group) and in 53% of patients with low MR-T2* (vs 22% in the high MR-T2*).

Previous studies investigated QT parameters in healthy people and TM patients [25, 30, 31]. Demircan et al. and Kayrak et al. reported prolonged P waves, QT and T peak-to-end dispersions, T peak-to-end intervals, and increased T peak-to-end/QT ratios, regardless of MR-T2* status [31, 32]. In addition, Najib et al. reported prolonged QTc interval in subjects with MR-T2* values < 20 ms [25]. Our results were in general agreement with reports by Demircan et al. and Kayrak et al., who showed no relationship between QT parameters and MR-T2* and ferritin level, except for QTc dispersion, which was prolonged in the MR-T2* <20ms group [31, 32].

To the best of our knowledge, studies on HRV assessment in TM patients have been limited. HRV shows changes of autonomic activity and its impact on heart function. TM patients with chronic anemia may have reduced HRV, which subsequently lead to tachycardia and a sustained decrease of autonomic fluctuations [33, 34]. Decreased HRV, even when patients show no symptoms, correlates with regional wall motion abnormality [35].

In our study, there were no statistically significant differences in all HRV parameters between ferritin groups and between MR-T2* groups. However, a comparison to HRV data based on age in normal subjects revealed that the HRV of our TM patients was below the normal range [14]. Heart rate deceleration capacity in 24 hours is a quantitative evaluation of vagal tone intensity. Reduced deceleration capacity means reduced vagal excitability, which correspondingly downgrades its protection of the human body and increases sudden cardiac death risk [36]. Our low MR-T2* patients and high ferritin level patients had lower deceleration capacity. Multivariate analysis also revealed an inverse correlation between heart rate deceleration capacity and MR-T2* values.

SAECG has been used to assess ventricular tachycardia as the underlying substrate in up to 5% of sudden cardiac deaths among TM patients. VLP measurement is a tool to assess for the presence of low voltage areas in the myocardium, which are mainly caused by infarction. In thalassemia patients, infarction may occur due to chronic anemia and iron deposits in the myocardium. The prevalence of VLP among the TM population varies between 3–31% [37, 38].

Contrary to those studies, none of our patients had detectable VLP using SAECG. We also found no correlation of myocardial iron concentration, as reflected by MR-T2*, with the

existence of VLP based on SAECG. Late gadolinium enhancement CMR examination may be done to evaluate fibrotic areas in myocardial tissue, but it is very expensive, invasive, and has limited availability [39]. Based on this study, we cannot propose SAECG as a routine screen for early diagnosis of ventricular arrhythmia, however, a simple examination such as electrocardiography to determine prolonged QTc interval and QTc dispersion might be sensitive enough to predict early ventricular arrhythmia.

There were several limitations to this study. First, MR-T2$^*$ examination was not performed in conjunction with the electrocardiography study. There was a one-year gap between MR-T2$^*$ examination and the start of this study. We did not repeat MR-T2$^*$ due to limited funds. However, according to previous studies, MR- T2$^*$ data would not differ much within a year [40, 41]. Second, the number of participants in the ferritin groups were not balanced, as it was difficult to find subjects with ferritin levels <2500 ng/ml because of their compliance in taking chelating drugs. As such, medication adherence needs to be monitored. A prospective cohort study is needed for further investigation.

## Conclusions

In our study, prolonged QTc dispersion and decreased heart rate deceleration capacity correlate with low MR- T2$^*$ as a reliable marker for myocardial iron deposition.

## Supporting information

**S1 Table. Demographic characteristics of patients with thalassemia major.**
(DOCX)

**S2 Table. Conduction patterns in the ferritin and MR-T2$^*$.**
(DOCX)

**S3 Table. Heart rate variability performance in the ferritin and MR-T2$^*$ groups.**
(DOCX)

**S4 Table. SAECG measurements in the ferritin and MR-T2$^*$ groups.**
(DOCX)

**S5 Table. Multivariate analysis between MR-T2$^*$ values and electrophysiological properties.**
(DOCX)

**S1 Dataset.**
(XLSX)

## Author Contributions

**Conceptualization:** Rubiana Sukardi, Pustika Amalia Wahidiyat, Mokhammad Ikhsan, Muhammad Yamin, Simon Salim, Mulyadi M. Djer.

**Data curation:** Phebe Anggita Gultom, Muhammad Yamin, Simon Salim.

**Funding acquisition:** Pustika Amalia Wahidiyat.

**Investigation:** Rubiana Sukardi, Phebe Anggita Gultom.

**Methodology:** Phebe Anggita Gultom.

**Project administration:** Pustika Amalia Wahidiyat, Mokhammad Ikhsan, Simon Salim, Mulyadi M. Djer.

**Resources:** Pustika Amalia Wahidiyat, Mokhammad Ikhsan, Muhammad Yamin, Mulyadi M. Djer.

**Software:** Rubiana Sukardi.

**Supervision:** Rubiana Sukardi, Pustika Amalia Wahidiyat, Mokhammad Ikhsan, Muhammad Yamin, Simon Salim, Mulyadi M. Djer.

**Validation:** Rubiana Sukardi, Pustika Amalia Wahidiyat, Mokhammad Ikhsan, Muhammad Yamin, Simon Salim, Mulyadi M. Djer.

**Writing – original draft:** Rubiana Sukardi, Phebe Anggita Gultom.

**Writing – review & editing:** Rubiana Sukardi, Phebe Anggita Gultom.

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
