## [Decision Letter · Decision Letter 0]

28 Jul 2022

PONE-D-22-13754Electrophysiological Properties and Heart Rate Variability of Patients with Thalassemia Major in Jakarta, Indonesia PLOS ONE

Dear Dr. Sukardi,

Thank you for submitting your manuscript to PLOS ONE. After careful consideration, we feel that it has merit but does not fully meet PLOS ONE’s publication criteria as it currently stands. Therefore, we invite you to submit a revised version of the manuscript that addresses the points raised during the review process.

Please address all comments indicated by the Reviewers.

We look forward to receiving your revised manuscript.

Kind regards,

Elena G. Tolkacheva, PhD

Academic Editor

PLOS ONE

Journal Requirements:

Reviewers' comments:

Reviewer's Responses to Questions

**Comments to the Author**

1. Is the manuscript technically sound, and do the data support the conclusions?

Reviewer #1: Yes

Reviewer #2: Partly

2. Has the statistical analysis been performed appropriately and rigorously? 

Reviewer #1: Yes

Reviewer #2: No

3. Have the authors made all data underlying the findings in their manuscript fully available?

Reviewer #1: Yes

Reviewer #2: No

4. Is the manuscript presented in an intelligible fashion and written in standard English?

Reviewer #1: Yes

Reviewer #2: No

5. Review Comments to the Author

Reviewer #1: The authors performed clinical investigations in beta-thalassemia patients with iron overload and accessed their cardiac functions using T2 star MRI, EKG, and HRV techniques. I suggest the authors include values of other iron parameters (such as serum iron and total iron-binding capacity) and numbers of blood transfusion units in the demographic data table.

Reviewer #2: The authors evaluated the association of electrophysiological parameters with serum ferritin levels and cardiac T2* values.

The manuscript has several issues that should be amended.

- In the Introduction, you stated that ventricular arrhythmia is one of the commonest electrophysiological abnormalities. In our experience supraventricular arrhythmias are more common than ventricular arrhythmias.

- It is well known that, besides iron levels, other factors can affect serum ferritin measurements (i.e. inflammation). Therefore, you should consider the mean value of serum ferritin levels in the year rather than a single measurement.

- There is no description of cardiac T2* quantification. Please, add it.

- Generally, not-normally distributed data are presented as median and Interquartile range and not as median and minimum and maximum. Please, correct the paper accordingly.

- Use “normal distribution” instead of “homogeneous distribution”.

- The considered patients are not 62. Please correct the text and report the demographic and clinical data of patients effectively considered.

- What do you mean with a “not clinically important difference”?

- You should perform a ROC curve analysis to evaluate if there is a QTc dispersion value that can predict iron overload with acceptable sensitivity and specificity.

- Which type of multivariate analysis did you perform? Please clarify. Where are the results of the univariate analysis?

- Any data about dose and frequency of chelators? At least partial information about compliance? You have really many patients with increased serum ferritin levels, but also a frequency of myocardial iron of 30% is not low……

- The Discussion needs to be shortened.

6. PLOS authors have the option to publish the peer review history of their article (what does this mean?). If published, this will include your full peer review and any attached files.

Reviewer #1: No

Reviewer #2: No

---

## [Author Response · Author response to Decision Letter 0]

11 Dec 2022

Thank you for the comments, we have revised our manuscript accordingly.

---

## [Decision Letter · Decision Letter 1]

28 Dec 2022

Electrophysiological Properties and Heart Rate Variability of Patients with Thalassemia Major in Jakarta, Indonesia

PONE-D-22-13754R1

Dear Dr. Sukardi,

We’re pleased to inform you that your manuscript has been judged scientifically suitable for publication and will be formally accepted for publication once it meets all outstanding technical requirements.

Kind regards,

Elena G. Tolkacheva, PhD

Academic Editor

PLOS ONE

Additional Editor Comments (optional):

Reviewers' comments:

Reviewer's Responses to Questions

**Comments to the Author**

1. If the authors have adequately addressed your comments raised in a previous round of review and you feel that this manuscript is now acceptable for publication, you may indicate that here to bypass the “Comments to the Author” section, enter your conflict of interest statement in the “Confidential to Editor” section, and submit your "Accept" recommendation.

Reviewer #2: All comments have been addressed

2. Is the manuscript technically sound, and do the data support the conclusions?

Reviewer #2: Yes

3. Has the statistical analysis been performed appropriately and rigorously? 

Reviewer #2: Yes

4. Have the authors made all data underlying the findings in their manuscript fully available?

Reviewer #2: Yes

5. Is the manuscript presented in an intelligible fashion and written in standard English?

Reviewer #2: Yes

6. Review Comments to the Author

Reviewer #2: I have no additional comments. The authors have addressed all my issues and significantly improved the manuscript.

7. PLOS authors have the option to publish the peer review history of their article (what does this mean?). If published, this will include your full peer review and any attached files.

Reviewer #2: No

---

## [Editor Report · Acceptance letter]

5 Jan 2023

PONE-D-22-13754R1 

Electrophysiological properties and heart rate variability of patients with thalassemia major in Jakarta, Indonesia 

Dear Dr. Sukardi:

I'm pleased to inform you that your manuscript has been deemed suitable for publication in PLOS ONE. Congratulations! Your manuscript is now with our production department. 

Kind regards, 

on behalf of

Dr. Elena G. Tolkacheva 

Academic Editor

PLOS ONE